# AIRAQc: Pre-Analytical Tool for Accurate Identification and Quantification of Artefacts in Histopathology

**Kuldeep Gautam**                                    KULDEEP.GAUTAM@AIRAMATRIX.COM
**Geetank Raipuria**                                  GEETANK.RAIPURIA@AIRAMATRIX.COM
**Nitin Singhal**                                       NITIN.SINGHAL@AIRAMATRIX.COM
*AIRA MATRIX, MUMBAI, INDIA*

**Editors:** Accepted for publication at MIDL 2025

## Abstract

Quality control in digitized Whole Slide Images (WSI) is critical to eliminate the impact of slide processing and scanning artefacts. Current artefact detection methods often lack comprehensive categorization or analyze images at a single magnification, limiting their efficacy. AIRAQc addresses these limitations through a multi-magnification approach, training on a large-scale dataset of over 40000 WSI from diverse sources. The solution yields superior performance as compared to existing approaches, testing on common artefacts types - air bubble, fold, pen mark and out-of-focus, and low inference time of 0.11 sec/ $mm^2$ of tissue.

**Keywords:** Artefact Detection, Histopathology, Quality Control

## 1. Introduction

Digital Pathology has seen widespread adoption in clinical and nonclinical applications. Tissue slides are scanned into high-resolution Whole Slide Images (WSIs), allowing for automated analysis using computational models and remote evaluation by pathologists. However, often pathologists encounter artefacts in the slides causing alterations in the tissue (Taqi et al., 2018). The presence of artefacts has also been shown to negatively impact the performance of downstream deep learning models (Wang et al., 2021; Raipuria and Singhal, 2022; Patil et al., 2023b; Schömig-Markiefka et al., 2021). Artefacts can be introduced during different stage of a digital whole slide preparation, including tissue sectioning, tissue fixation, staininig, and scanning. Artefacts on diagnostically important tissue might require re-scanning or re-processing, which could cause delays. The type of artefact introduced and the proportion of tissue involved determine the extent of the delay.

Therefore, it is essential to detect artefacts before the WSI reaches the pathologist, as this enhances quality control (QC) and helps minimize delays in reporting. Deep learning based models are apt for the tasks of artefact detection as they can process large volumes of images. A practical artefact detection solution should be designed with a couple of key features. In order to identify the slide for re-processing or re-scanning, it must first accurately detect and classify various types of artefacts, such as air bubbles, tissue folds or out-of-focus regions. Secondly, it should have a fast inference time to accommodate the high volume of tissue specimens that are typically processed in a pathology workflow.

Current methodologies for histopathological artefact detection face two key limitations: (1) insufficient systematic classification of artefacts (Janowczyk et al., 2019; Patil et al., 2023a), (2) reliance on single-magnification image analysis (Weng et al., 2024; Smit et al.,

| | Model(s) | Magnification | Artefact Types |
|---|---|---|---|
| (Smit et al., 2021) | Tissue Segmentation + DeepLabV3+ with EfficientNet B2 | 2.5x | Air Bubble, Pen Mark, Fold, Out-of-Focus, Ink, Dust |
| **GrandQC** (Weng et al., 2024) | UNet++ with EfficientNetB0 tissue detection and artefact detection | 5x or 7x or 10x | Air Bubble, Pen Mark, Fold, Out-of-Focus, DarkSpot |
| (Schreiber et al., 2024) | Otsu Thresholding | Low Magnification | Pen Mark |
| **HistoQC** (Janowczyk et al., 2019) | Image Processing | 40x | No Categorization |
| **HistoROI** (Patil et al., 2023a) | Patch Classifier (ResNet-18) | 20x or 40x | Epithelium, Stroma, Lymphocytes, Adipose, artefacts, and Miscellaneous |
| **PathProfiler** (Haghighat et al., 2022) | Tissue Segmentation + Multi-Label ResNet-18 | 5x | Fold, Out-of-focus, Staining Issues, and Miscellaneous |
| **AIRAQc** | Multi-Mag Transformer for tissue and artefact detection | Multi-Mag 5x & 10x | Air Bubble, Pen Mark, Dark Spot, Fold, Out-of-Focus, Knife Line Coverslip, Missing Tissue |

Table 1: Overview of various solutions for artefact detection in digital Histopathology

2021; Patil et al., 2023a; Haghighat et al., 2022). These shortcomings hinder their adaptability to diverse laboratory workflows and multi-scale diagnostic requirements. Table 1 provides an overview of existing QC solutions. In this study, we present a multi-magnification approach to artefact detection, classification and quantification. The WSI images are analysed at different magnifications to identify specific artefacts without losing context or fine-grained information. The algorithm specifically quantifies artefacts within the tissue region while disregarding non-essential artefacts outside of it.

To evaluate multi-magnification benefits in artefact detection, we manually annotated 50 H&E stained WSIs from The Cancer Genome Atlas (TCGA) dataset (Weinstein et al., 2013). Comparative analysis against existing methods (Patil et al., 2023a; Weng et al., 2024; Janowczyk et al., 2019) demonstrates that AIRAQc outperforms prior approaches by leveraging resolution-specific artefact identification. The framework's advantage stems from its ability to detect distinct artefacts more effectively across magnification levels. Annotations for the TCGA test set are publicly released to facilitate reproducibility and community benchmarking on the following https://tinyurl.com/qctcga.

## 2. Method

Artefacts exhibit varying impacts on tissue visibility, depending on magnification. For example, blur may not be visible at lower magnifications where a pen marks are easily identifiable. Air bubbles and large folds can appear as out-of-focus tissue at higher magnifications. To address this complexity, we train a multi-magnification model with data from three different magnifications - 2.5x, 5x, & 10x with concentric field-of-view. The multi-magnification images are processed through transformer attention layers of an encoder module that shares weights across magnifications, followed by fusing of the multi-magnification features by taking crops of features corresponding to the lower magnification inputs. A light weight MLP decoder is employed to produce segmentation outputs. For detailed architecture specifics,

| Method | Mag. | Out-of-Focus | Fold | pen mark | air bubble | Binary |
|--------|------|--------------|------|----------|------------|--------|
| **HistoQC** | 40x | - | - | - | - | 0.60 |
| **HistoROI** | 20x/40x | - | - | - | - | 0.68 |
| **GrandQC** | 10x | 0.70 | 0.24 | 0.83 | 0.81 | 0.83 |
| **AIRAQc** | 2.5x+5x+10x | 0.92 | 0.84 | 0.91 | 0.90 | 0.92 |

Table 2: Pixel-level Dice score for detection and classification of artefacts

readers are referred to prior work (Raipuria et al., 2023). Additionally, a separate decoder is trained for tissue region segmentation using the same encoder. The two models are trained on annotated data obtained from 40000+ WSI from numerous organs, species, scanners, and laboratories, including H&E and IHC stains.

## 3. Experimental Setup and Results

We compare AIRAQC with GrandQC, HistoROI and HistoQC. Two metrics are used for the evaluation, a Multi-class Dice score on tissue affected by a) air bubble, b) fold, c) pen mark and d) out-of-focus artefacts; binary Dice score for total artefact affected tissue area. AIRAQc, GrandQC and PathProfiler produce multiclass artefact masks; whereas HitoQC generates only a binary prediction for tissue affected by artefacts. All masks are intersected with the tissue mask to exclude off-tissue regions as the artefacts on non-tissue area do not hinder manual or automated analysis.

Test Dataset includes 50 WSIs, manually annotated with multiclass artefact masks, from Colorectal, Breast, Lung, Stomach, and Cervical Cancer tissues[1]. The test set comprises 25 WSIs containing artefacts and 25 artefact-free WSIs. All artefacts are annotated across a range of magnifications, from 2.5x to 10x.

Quantitative results are presented in Table 4. Propelled by diverse dataset used for training, multi-magnification inputs and transformer based architecture, AIRAQc out performs all other solutions. As compared to GrandQC, the highest gain it observed for out-of-focus artefact. Figure 1 provides qualitative results for each artefact. It can be observed that other methods are unable to identify the tissue affect by artefacts or mis-classify them. Furthermore, AIRAQc processes a $mm^2$ of tissue in 0.11 sec when running with 2.5x + 5x + 10x magnification, on NVIDIA V100 GPU (32GB) and Intel(R) Xeon(R) W-2145 CPU (3.70GHz).

## 4. Conclusion

Our findings demonstrate that AIRAQc improves the reliability of artefact detection and classification as compared to existing solutions, and can streamline laboratory operations by rapidly inferring on large volume of data, ultimately supporting pathologists delivering timely and accurate observations.

---

1. The test WSI are not included in the training data

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

## Appendix A. Qualitative Results

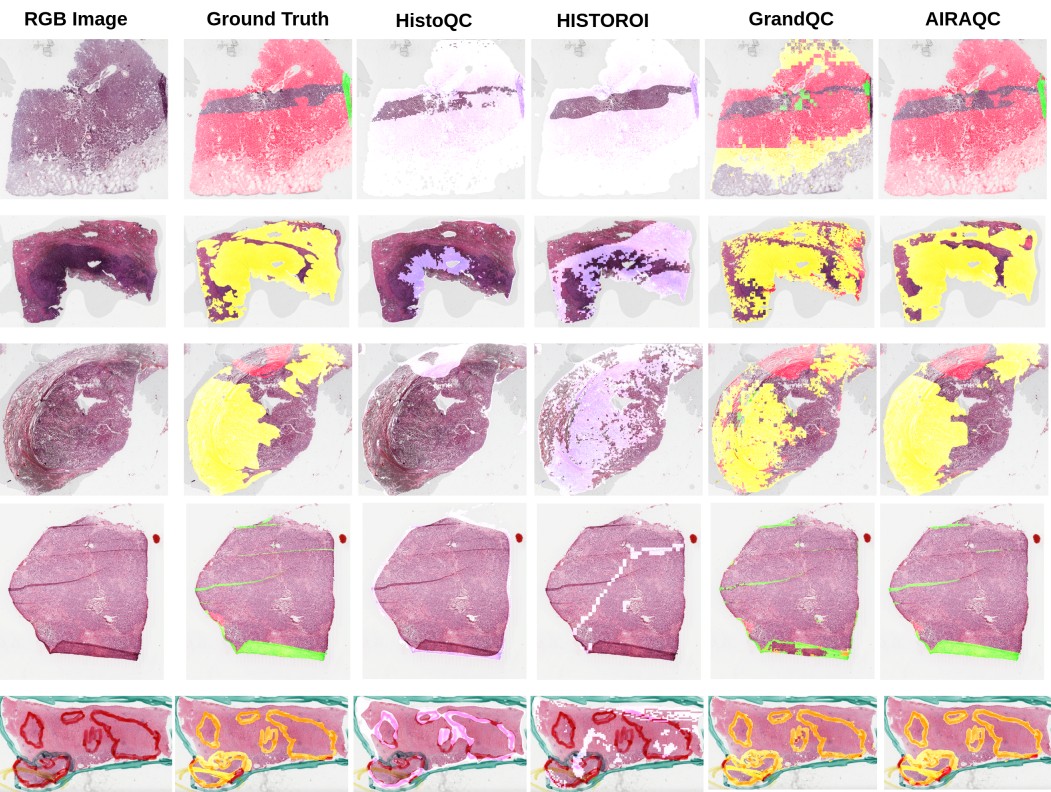

Figure 1: Figure provides qualitative comparison for HistoQC, HistoROI, GrandQC and AIRAQC. Color map for the overlay - out-of-focus: red, air-bubble: yellow, fold: green, penmark:orange, binary mask: white

