# OpenReview forum: "AIRAQc: Pre-Analytical Tool for Accurate Identification and Quantification of Artefacts in  Histopathology"
_MIDL.io/2025/Short_Papers — MIDL 2025 - Short Papers_

### Official Review · Reviewer_2iaA · 2025-04-28

**Rating:** 4
**Confidence:** 4

**Summary:**

In this paper, a multi- magnification approach to whole slide imaging (WSI) artefact detection, classification and quantification is proposed. The authors also provide a dataset of 50 WSI from a TCGA dataset.

**Strengths:**

1. A framework to detect artifacts in WSI has been proposed using a transformer encoder that fuses information from  multi-magnifications and a MLP-based decoder to segment the regions containing artifacts.
2. The authors also provide a dataset of 50 WSI from a TCGA dataset.
3. The performance seems better than other approaches on the same dataset.

**Weaknesses:**

1. It is unclear what dataset AIRAQc was trained on.
2. It is not clear if it was trained and tested on the same dataset, so comparison with other approaches on this dataset (out-of-distribution) may not be entirely an apples-to-apples comparison if it was trained on the same dataset.

---

### Decision · Program_Chairs · 2025-05-01

Accept